# Promoted Mid-Infrared Photodetection of PbSe Film by Iodine Sensitization Based on Chemical Bath Deposition

**DOI:** 10.3390/nano12091391

**Published:** 2022-04-19

**Authors:** Silu Peng, Haojie Li, Chaoyi Zhang, Jiayue Han, Xingchao Zhang, Hongxi Zhou, Xianchao Liu, Jun Wang

**Affiliations:** 1School of Optoelectronic Science and Engineering, University of Electronic Science and Technology of China, Chengdu 610054, China; pengsilu0505@163.com (S.P.); lhj_uestc@163.com (H.L.); zhangcy1009@sina.com (C.Z.); hanjiahue_uestc@163.com (J.H.); azhangxingchao@163.com (X.Z.); liuxc@uestc.edu.cn (X.L.); 2State Key Laboratory of Electronic Thin Films and Integrated Devices, University of Electronic Science and Technology of China, Chengdu 610054, China

**Keywords:** chemical bath deposition, PbSe, iodine sensitization, photodetector

## Abstract

In recent years, lead selenide (PbSe) has gained considerable attention for its potential applications in optoelectronic devices. However, there are still some challenges in realizing mid-infrared detection applications with single PbSe film at room temperature. In this paper, we use a chemical bath deposition method to deposit PbSe thin films by varying deposition time. The effects of the deposition time on the structure, morphology, and optical absorption of the deposited PbSe films were investigated by x-ray diffraction, scanning electron microscopy, and infrared spectrometer. In addition, in order to activate the mid-infrared detection capability of PbSe, we explored its application in infrared photodetection by improving its crystalline quality and photoconductivity and reducing tge noise and high dark current of PbSe thin films through subsequent iodine treatment. The iodine sensitization PbSe film showed superior photoelectric properties compared to the untreated sample, which exhibited the maximum of responsiveness, which is 30.27 A/W at 808 nm, and activated its detection ability in the mid-infrared (5000 nm) by introducing PbI_2_, increasing the barrier height of the crystallite boundary and carrier lifetimes. This facile synthesis strategy and the sensitization treatment process provide a potential experimental scheme for the simple, rapid, low-cost, and efficient fabrication of large-area infrared PbSe devices.

## 1. Introduction

Binary lead chalcogenides (PbS, PbSe, PbTe) are of great technical significance due to their application as mid-wavelength infrared detectors near room temperature [1,2,3,4]. Among these lead chalcogenides, PbSe is a typical direct band gap semiconductor with a narrow bandgap of 0.27 eV at room temperature. Due to its excellent optical, electrical, optoelectronic, and chemical stability properties, as well as the simplicity of thin film preparation, PbSe is an attractive candidate for infrared photodetection [5,6,7,8,9,10], solar cells [11,12,13], light emitting devices [14,15,16], and thermoelectric applications [17,18,19,20,21]. The optoelectronic properties of lead chalcogenide films have been the subject of extensive research due to the broad spectral detection in the mid-infrared (3–5 µm) wavelengths at room temperature [22,23,24]. Currently, the deposition techniques are most widely used for the deposition of photo-responsive PbSe thin films, including vacuum-based thermal evaporation [22], molecular beam epitaxial growth (MBE) [25,26,27], pulsed laser deposition (PLD) [28], magnetron sputtering [29,30], and chemical bath deposition (CBD) [31,32]. The CBD method is a low-cost process compared to the preparation methods mentioned above, and the quality of the obtained films is comparable to that of other films obtained with more complex deposition processes, which have recently emerged as the method for metal chalcogenide film deposition [32,33,34]. CBD methods are currently attracting considerable attention because they only require ordinary magnetic water bath stirrers instead of large and expensive equipment such as vacuum systems. In addition, the starting chemicals in the synthesis reaction are commonly available and cheap. The reaction typically occurs in a low-temperature aqueous solution and can rapidly deposit large-area thin films on substrates through the CBD method. However, PbSe thin-film photodetector made by CBD is usually affected by a high dark current owing to its narrow band gap and thermally exciting carriers at room temperature. The photoresponsivity and detection rate of PbSe films can be effectively enhanced by doping some deep-level elements (oxygen or iodine) and recrystallization, called sensitization, enabling them to work at room temperature [35,36]. The optoelectronic properties of sensitized PbSe films are usually affected by crystal structure, crystallite size, and stoichiometry. Sensitization treatment results in the growth of lead iodides (PbI_2_), lead oxides (PbO_2_), and/or lead-oxygen-selenium compounds (such as PbSeO_3_ and PbSeO_5_) along the grain boundaries of PbSe [22,32,37,38].

Thus, in a sensitized polycrystalline film, each grain is effectively enveloped or passivated with an oxide or halide (depending on sensitization conditions), which is a wide-bandgap material. This passivation constitutes the formation of a heterojunction at each grain boundary, resulting in the separation of photo-generated carriers with holes migrating to the interface material regions and electrons getting trapped in the bulk grains. This process is believed to be responsible for carrier inversion and an increase in carrier lifetimes in the sensitized films [37,39]. Therefore, each grain is effectively passivated or enveloped by an oxide or iodide with a wide-bandgap material in the sensitized film. This passivation forms a heterojunction at each grain boundary, resulting in the separation of photogenerated carriers, electrons becoming trapped in the bulk grains, and the migration of holes to the interfacial material region, which lead to an increase in the carrier lifetime of the sensitized film.

For now, most of the studies only report on the PbSe films obtained by CBD, studying the effects of temperature, time, pH, and other factors on the grown films. Another part of the study reported the effects of oxidation and iodination on the electro-optical properties of thin films. However, there are few reports combining a CBD synthesis process, iodine sensitization treatment, optoelectronic performance testing, and an infrared gain mechanism. This work provides a detailed analysis from several perspectives, including experimental synthesis, subsequent processing, performance testing, and mechanism interpretation, which are lacking in previous research reports.

Therefore, in this work, we use a facile and convenient CBD growth strategy to synthesize large-area PbSe films composed of PbSe directly on amorphous glass substrates. By controlling the deposition time to design the growth of PbSe thin films, the films with different thicknesses and different crystal sizes are obtained, and we achieved the controllable growth of the films. Through the XRD, SEM, and FTIR characteristics, it is proved that the synthesized films are dense with the Fm-3m space group of cubic crystal systems, and that there is an obvious absorption peak in the near infrared to far infrared. In order to improve the photoelectric response and reduce the noise of the high dark current of the PbSe film device, we used a unique post-deposition sensitization step, which is annealing the film in an iodine atmosphere, so as to explore its use for the low-cost fabrication of near-infrared photodetectors. The measured near-infrared photo-response of PbSe after iodination exhibited corresponding responsiveness (abbreviated as R_i_) at 808 and 1550 nm, in which the maximum of R_i_ is 30.27 A/W at 808 nm with a bias of 1 V under incident power density of 233 µW/cm^2^ while 1.75 A/W at 1 V is under 1550 nm with an incident power density of 198 µW/cm^2^, indicating the excellent photodetection capabilities of the device. Moreover, the iodinated PbSe can also achieve a mid-infrared spectrum detection capability ranging from 2250 nm–5000 nm. The results reveal the critical role of iodine sensitization in enhancing the photocurrent of PbSe materials and activating its detection ability in the mid-infrared, which is ameliorated by increasing the barrier height of the crystallite boundary and carrier lifetimes in the sensitized films.

## 2. Materials and Methods

### 2.1. Synthesis of PbSe Films

PbSe films were synthesized by the CBD method following the method reported by Hemati. et al. [31] with modifications. In order to deposit the PbSe film on a glass substrate, two precursors of Pb cations and Se anions should be prepared. The synthesis method of the Se precursor (Na_2_SeSO_3_) is reported as in the previous work [31], and the specific synthesis steps are shown in Appendix A. In this work, we used 2.4375 g lead (II) acetate trihydrate (Aladdin, 99.99% trace metals basis) as the Pb precursor. Additionally, sodium hydroxide (NaOH) was used as an agent to adjust the pH of the reaction solution. Rectangular microscope slides are used as substrates, which are washed with alcohol, acetone solvent, and deionized water for 15 min, respectively, and then dried with nitrogen for use. To prepare the deposition solution, first, the Pb (CH_3_COO)_2_ (2.4375g) was added into the 50 mL deionized water under stirring to dissolve and was then heated to reach 60 °C. Then, NaOH was added to control the pH of the solution reach to 10. Before adding Na_2_SeSO_3_ solution, the glass substrate was put in the beaker at a 70° angle relative to the air-solution interface. Finally, 50 mL of Na_2_SeSO_3_ was added to the beaker, and the PbSe film was deposited for 10 min, 20 min, and 30 min (named as PbSe-T10, PbSe-T20, PbSe-T30, respectively). It is worth mentioning that the reaction solution is constantly stirred during the preparation process. Figure 1 illustrates the schematic diagram of the obtained PbSe thin films on glass. The PbSe films were prepared by simple CBD at different deposition times, which obtained different micro morphology and appearance color of the films.

In order to further improve the photoelectric response, reducing the noise and high dark current of the PbSe films detector, a unique post-sensitization treatment is required, namely annealing the film at high temperature in a certain atmosphere (oxygen gas or and iodine atmosphere). S. Ganguly et al. discussed in detail the carrier transport model in sensitized PbSe compounds [40]. The iodine transport in the sensitization process depends on the morphology, crystal structure, and grain size of the lead salt film, which is more effective in thin films with smaller grains. Among the films prepared above, PbSe-T30 not only has a higher photocurrent, but also has a smaller particle size for sensitization (seen from photocurrent diagram the SEM figure). Therefore, in the next process, we chose PbSe-T30 for subsequent sensitization, that is, annealing the film in an iodine vapor atmosphere to study its photoelectric properties. The PbSe-T30 samples were annealed in the iodine atmosphere for 20 min at 200 °C (The sample is named as I_2_-PbSe-T30). After iodization treatment, the grain surface can be passivated to optimize its crystal phase structure.

### 2.2. Materials Characterization

Powder XRD patterns of the obtained PbSe films were performed using an X-ray diffractometer (Rigaku Ultima IV, Tokyo, Japan) with Cu Kα radiation. The scanning electron microscope (SEM) used a field emission scanning electron microscope (Geminisem 300, Jena, Germany) to observe the microscopic topography of the PbSe films. We used a Fourier transform infrared spectrometer (FTIR-Nicolet iS50, Waltham, MA USA) to test the optical properties of the PbSe films. X-ray photoelectron spectroscopy (XPS, XSAM800, Manchester, UK) was used to confirm the element valence of the PbSe-T30 and I_2_-PbSe-T30. An Ultraviolet Photoelectron Spectrometer (UPS, Thermo Fisher ESCALAB 250Xi, Waltham, MA USA) was carried out to obtain the energy band alignment of the PbSe-T30 and I_2_-PbSe-T30. The PbSe film detectors were irradiated with commercial small semiconductor lasers of various wavelength bands, and their photoelectric response properties were measured by a source instrumentation system (Keithley 2636B Tektronix Inc, Beaverton, OR USA). The noise power density of the PbSe devices was measured by a PDA (Tektronix Inc, Beaverton, OR USA), which is a semiconductor parameter analyzer.

### 2.3. Device Fabrication

The Au electrodes with a thickness of 120 nm of PbSe thin films were fabricated under a vacuum of 4 × 10^−4^ Pa using ZD-400 thermal deposition equipment. The prepared PbSe device had a channel size of 100 µm × 50 µm.

## 3. Results and Discussion

### 3.1. Structural Data

XRD patterns were measured to study whether the synthesized thin film prepared in the experiment was PbSe. Figure 2a showed typical XRD patterns of PbSe thin films under different deposition times and I_2_-PbSe-T30, respectively. In the XRD patterns, there are obvious peaks at 2θ = 25°, 27°, 41.7° and so on, indicating that the XRD pattern of the samples can be completely indexed in a cubic phase of Fm-3m, which is a standard card (JCPDF No. 65-0347) of PbSe. From the XRD analysis, it can be seen that the obtained I_2_-PbSe-T30 is multiphase in structure, and two phases are presented in the diffraction pattern. One phase is cubic phase PbSe, the other phase can be indexed to a trigonal phase of P-3m1, which is a standard card (JCPDF No. 07-0235) of PbI_2_. To further confirm the crystal structure and unit cell parameters of the as-prepared PbSe films, Rietveld refinement was performed on the XRD patterns of the PbSe films using GSAS software. The cross symbol represents the experimental XRD pattern, and the solid line represents the fitted XRD pattern. The fitting results are shown in Figure 2b–d. As can be seen, the measured and fitted XRD patterns match very well, indicating that the analysis of the crystal structure should be reasonable. Additionally, the detailed refinement results can be found in Appendix A.

Figure 3a shows the crystal structural diagram of PbSe, which has a rock salt structure and belongs to the cubic of Fm-3m space group. The grey atoms represent lead, and the red atoms represent selenium. Figure 3b shows the crystal structural diagram of PbI_2_ which belongs to the trigonal phase of the P-3m1 space group. The grey atoms represent lead, and the purple atoms represent iodine.

In order to investigate the optical properties of the prepared PbSe films, the near infrared to middle infrared absorption spectra were characterized. From the light absorption spectrum Figure 4a,b, it can be seen that PbSe films have broad light absorption in the range of 1000 to 25,000 nm. In addition, the sample has obvious characteristic absorption peaks at 1400, 1900 nm and in the range of 10,000 to 15,000 nm, which are consistent with the previously reported literature results [41,42]. Besides, as the deposited time increases, the absorbance value of PbSe gradually increases. Moreover, after iodization, the I_2_-PbSe-T30 has a higher absorbance value than the PbSe-T30, indicating that iodination can indeed increase the absorption of the PbSe.

Figure 5 shows top view SEM images of the PbSe-T10, PbSe-T20, PbSe-T30 and I_2_-PbSe-T30. The SEM observation shows that with an increase in deposition time, the grain size decreases. The average crystallite size decreased from 100 nm to 20 nm as the deposition time was varied from 10 min to 30 min. Figure 5a shows the morphology of PbSe-T10, which is made of large grains. From the Figure 5b, we can see that the large particles in the PbSe-T20 film have been partially transformed into nano-spheres. It converts from the large grains-shape to the nano-spheres-shape. Figure 5c displays the morphology of the PbSe-T30. This film is fully characterized by the small spherical nanocrystals, which is totally different from PbSe-T10. The mechanism that causes the different particle sizes of the film under different deposition times may be the mode of film growth change from Ion-by-Ion (IBI) to cluster, which has been explained in detail in the previous reports [31]. Figure 5d shows the SEM image of I_2_-PbSe-T30 films. From Figure 5d, it can be seen that the I_2_-PbSe-T30 has better crystallinity and clearer grain boundaries, indicating that iodination annealing can indeed change the crystallinity of the material. Figure 5e–i is the mapping diagram of PbSe-T30 and I_2_-PbSe-T30. Pb and Se are evenly distributed in the PbSe-T30 film, and the three elements of Pb, Se, and I in the I_2_-PbSe-T30 are also evenly distributed in the film, indicating that the iodine element successfully entered the film after the iodination annealing process. Figure 5j,k show the EDS spectra of PbSe-T30 and I_2_-PbSe-T30, respectively. Figure 5j shows that there are Pb and Se peaks in PbSe-T30 and that Figure 5k is the EDS spectrum of I_2_-PbSe-T30. There are peaks of Pb, Se, and I in the figure, and through Table 1, we can intuitively see the content of each element in the two samples. In the sample of PbSe-T30, the average atomic percentage of Se and Pb in the sample is about 50%, which is in line with the stoichiometry of PbSe. In addition, the element content of the Pb, Se, and I of the I_2_-PbSe-T30 is 54.9%, 29.27%, and 15.83%, respectively. It can be seen that the Se content is significantly reduced. This may be due to the high annealing temperature causing part of Se to volatilize, which also indicates the presence of selenium vacancies. During sensitization, iodine atoms gradually diffused into a PbSe lattice and occupied those selenium vacancies [43].

As mentioned above, many characteristics of PbSe films can be changed by the sensitization treatment, so it is essential to investigate the chemical states of the elements contained in the PbSe-T30 and I_2_-PbSe-T30 using the XPS. The XPS spectra of the Pb, Se, and I elements before and after the sensitization are quite different, as shown in Figure 6a–f. For PbSe-T30, the Pb element (Figure 6a) is dominated by two peaks. The first doublet peak located at lower Binding Energy (abbreviated as BE) (at about 137.7 eV) is associated with 4f7/2 spin-orbit, and the second one located at higher BE (at about 143.0 eV) arises from 4f5/2 spin-orbit of Pb [44]. For the Pb element of I_2_-PbSe-T30, there are two chemical states, PbSe (approximately 138.2 eV and 143.0 eV) and PbI_2_ (approximately 139.2 eV and 143.9 eV), as shown in Figure 6b. Since the higher BE Pb peak in each doublet band is assumed to be bonded to I, they are suggested to be related to the combination of Pb-I. Further, due to the lower BE of PbSe compared to Pb-I, the lower-energy features (at about 137.3 eV and 142.3 eV) are assigned to Pb in PbSe. For the Se element, the Se transitions of 3d, corresponding to Se^2−^ in PbSe at 53.5 eV, and Se^4+^ in SeO_2_ at 58.7 eV were observed, respectively [45,46]. The oxygen content in the samples is very small and can be ignored, which is not observed in the XRD. Comparing Figure 6c,d, it can be seen that after iodination annealing, the Se in I_2_-PbSe-T30 volatilizes and is replaced by iodine, causing the peak intensity to drop sharply. This result also corresponds to the EDS. It can be seen from Figure 6e that there is no iodine in the PbSe-T30. For the I element of I_2_-PbSe-T30, as shown in Figure 6f, I 3d3/2 and 3d5/2 transition at 630.4 eV and 618.8 eV, corresponding to I_2_-in PbI_2_, indicating that the material is partially iodinated [36].

### 3.2. Photoelectric Properties

Plots of photocurrent versus time of PbSe-T10, PbSe-T20, PbSe-T30 and I_2_-PbSe-T30 with the laser turned on and off at an applied bias of 1V under 808 nm with an incident optical power of 1.18 Wcm^−2^ are shown in Figure 7a, and the illustration in Figure 7a is a schematic diagram of a PbSe photoconductive device. It can be seen that under the same test conditions, the PbSe-T30 exhibits a higher photocurrent than the PbSe-T10 and PbSe-T20. The reason why the performance of the PbSe-T30 sample is better than the two samples may be its different surface morphology, which is composed of small nanoparticles, and its thickness. In addition, Figure 7a also shows photocurrent response curves in the time domain of the I_2_-PbSe-T30 films. It is obvious that the photocurrent is significantly enhanced after the annealing sensitization treatment. Figure 7b shows the responsivity comparison of PbSe-T30 and I_2_-PbSe-T30. It can be seen that I_2_-PbSe-T30 has a higher responsivity under the same test conditions. In addition, PbSe-T30 has no response to 2250 nm, but I_2_-PbSe-T30 is responsive at 2250 nm with a responsivity of 0.237 A/W, which means that iodination can activate the detection capability of mid-infrared PbSe. Figure 7c shows the photocurrent curves of I_2_-PbSe-T30 at different incident optical powers from 84 mWcm^−2^ to 0.25 mWcm^−2^ and 69 mWcm^−2^ to 0.2 mWcm^−2^ at 1310 nm and 1550 nm, respectively. It can be seen that with decreasing power density, there is an overall decrease in photocurrent. Figure 7d shows a photocurrent on-off state at 808 nm for I_2_-PbSe-T30 at an incident optical power of 50 mWcm^−2^ and under 1 V. We can see that the rise time and decay time of the device are approximately 2.1 s and 2.3 s. Additionally, other photoelectric properties are shown in the Appendix A. The noise power density as a function of the frequency of PbSe-T30 and I_2_-PbSe-T30 was presented in Appendix A, in which the noise has been obviously optimized for PbSe-T30 after iodine treatment. A histogram of the photocurrent value of the I_2_-PbSe-T30 device from 808 nm to mid-infrared waveband (5000 nm) is shown in the Appendix A, indicating its detection capability in the mid-infrared band. What’s more, Appendix A shows the corresponding I–V curves under light-off conditions and illuminated by an 808–1550 nm light source. Plots of photocurrent versus time of I_2_-PbSe-T30 with the laser turned on and off under 1V from 808 to 1550 nm with different incident optical power are shown in Appendix A.

Based on the following formulas for photocurrent (I_ph_) and responsivity (R_i_), the corresponding performance parameters of PbSe films were further evaluated.
I_ph_ = /I_photo_−I_dark_/(1)
R_i_ = I_photo_/P_light_×A(2)
where I_ph_ is photocurrent, P_light_ is incident light power density and A represents the effective photosensitive area of the device. Figure 7e exhibits the relationship between the photocurrent and different optical power densities, and the curve relationship in the figure conforms to the well-known linear law. It can be described as the overall increase in photocurrent as the power density increases, because stronger light will generate more photogenerated carriers, which results in an increase in photocurrent. In addition, Figure 7f shows the R_i_ corresponding to I_2_-PbSe-T30. It can be seen that the corresponding power density decreases and the device’s responsivity increases. When the incident power density of I_2_-PbSe-T30 device is 233 µW/cm^2^, the maximum loudness of 808 nm measured at a bias voltage of 1 V is 30.27 A/W. When the incident power density is 198 µW/cm^2^ and the bias voltage is 1 V, the responsivity at 1550 nm is measured to be 1.75 A/W, indicating that the device has good photodetection capabilities. In addition, the responsivity at 5000 nm is measured to be 0.053 A/W, which demonstrates its good response in the mid-infrared band.

In order to further explain the mid-infrared response gain mechanism of I_2_-PbSe-T30, we performed UPS measurements to obtain the energy band alignment of the PbSe-T30 and I_2_-PbSe-T30 and analyzed the energy band structure, as shown in Figure 8a,b. The work functions of PbSe-T30 and I_2_-PbSe-T30 were determined to be 4.23 and 4.46 eV, respectively, by subtracting the second electron cutoff energy from the photo energy of the He I light source (21.1 eV), as shown in the Figure 8a [47]. The valence band of PbSe-T30 and I_2_-PbSe-T30 were 0.13 and 1.07 eV, which is close to the previously reported results of PbSe and sensitization PbSe. After iodination, I_2_-PbSe-T30 is a composite phase of PbI_2_ and PbSe, that is, the internal structure is PbI_2_/PbSe/PbI_2_, which can be proved from XRD and XPS. Figure 8b exhibits the energy band diagram of the internal structure of I_2_-PbSe-T30 under mid-infrared irradiation after applying 1 V. During mid-infrared irradiation of the device during the biasing, the PbI_2_ layer will not be able to generate electron-hole pairs due to its high bandgap energy. Conversely, electron-hole pairs will be generated in the PbSe layer by the MWIR illumination and separated by the applied bias. Due to the high barrier electron height at the PbI_2_ interface, electrons cannot flow through the device. Therefore, holes can pass through PbI_2_ and reach the electrode. The Figure 8b shows that due to the high electronic barrier, electrons cannot be injected from the electrode, and the electron trap is activated after the iodination process, which effectively prevents the recombination of holes and electrons, increases the life of carriers, and stimulates the infrared response of the PbSe.

In addition, the preparation methods and optoelectronic parameters of previous related PbSe phototransistors and the I_2_-PbSe-T30 device are summarized in Table 2. The response speed of PbSe thin-film detectors synthesized using CBD or the chemical solution method tends to be slow [48]. The speed of I_2_-PbSe-T30 is indeed improved compared to the detector prepared by the same method, and the detection band can be broadened to the mid-infrared, which is even better than the previous hybrid heterostructure.

## 4. Conclusions

To summarize, we have directly synthesized large-area PbSe films composed of crystal particles with different sizes on amorphous glass substrates using the CBD method by controlling deposition time. Moreover, through subsequent iodine treatment to improve the crystalline quality and photoconductivity and to reduce the noise and high dark current of PbSe films, we explored their use for infrared photodetection. The iodine sensitization sample showed superior photoelectric properties compared to the untreated sample, which exhibited that the maximum of R_i_ is 30.27 A/W at 808 nm with a bias of 1 V under an incident power density of 233 µW/cm^2^. Moreover, iodine sensitization can also activate its detection ability in the mid-infrared (5000 nm) by increasing the barrier height of the crystallite boundary and carrier lifetimes. This work provides a detailed analysis from several perspectives, including experimental synthesis, subsequent processing, performance testing, and mechanism interpretation, which are lacking in previous research reports. This facile synthesis strategy, combined with the sensitization treatment process, provide a potential experimental scheme for the simple, rapid, low-cost, and efficient fabrication of large-area infrared PbSe devices. Moreover, this iodination process can also be applied to binary chalcogenide lead to improve its photoelectric response.

## Figures and Tables

**Figure 1 nanomaterials-12-01391-f001:**
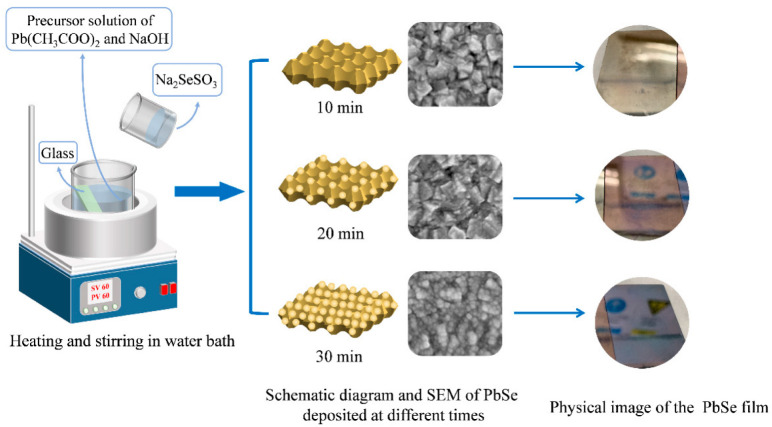
The experimental flow chart of PbSe thin film preparation and its SEM image and physical image.

**Figure 2 nanomaterials-12-01391-f002:**
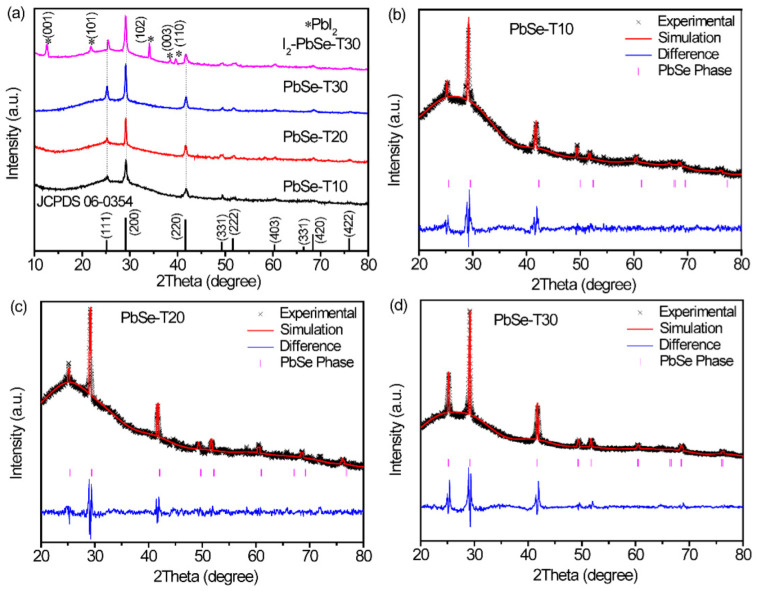
(**a**) XRD spectrum of the PbSe films which deposited for different times (10 min, 20 min, 30 min) and I2-PbSe-T30 films; The positions marked with * represent diffraction peaks corre-sponding to the PbI2 phase. Refinement results for PbSe: (**b**) PbSe-T10; (**c**) PbSe-T20; (**d**) PbSe-T30. The blue line represents the difference between the experimental and simulated results.

**Figure 3 nanomaterials-12-01391-f003:**
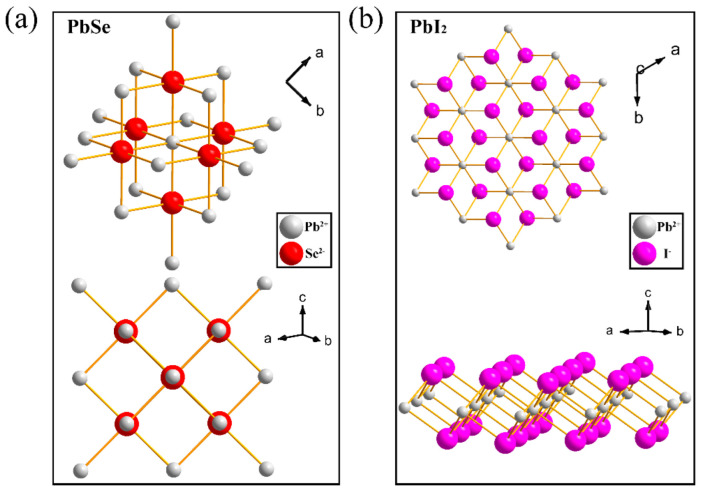
(**a**) Crystal structural diagram of PbSe, the grey atoms represent lead, and the red atoms represent selenium.; (**b**) Crystal structural diagram of PbI_2_., the grey atoms represent lead, and the purple atoms represent iodine.

**Figure 4 nanomaterials-12-01391-f004:**
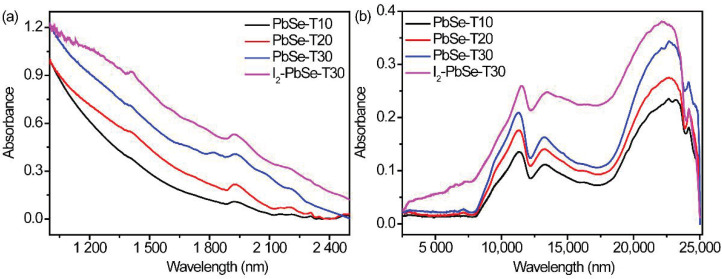
Optical absorption spectra of PbSe films: (**a**) Near-infrared absorption spectrum of PbSe films; (**b**) Mid-infrared absorption spectrum PbSe films.

**Figure 5 nanomaterials-12-01391-f005:**
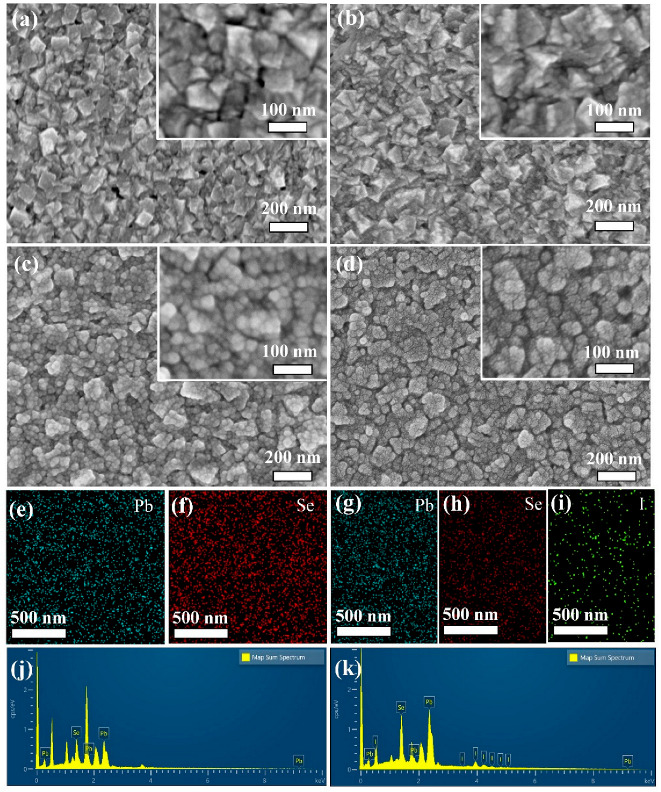
Top-view SEM images of the PbSe thin film deposited under different times: (**a**) PbSe-T10; (**b**) PbSe-T20; (**c**) PbSe-T30; (**d**) I_2_-PbSe-T30. EDS elemental maps of PbSe-T30: (**e**) Pb; (**f**) Se. And EDS elemental maps of I_2_-PbSe-T30: (**g**) Pb; (**h**) Se; (**i**) I. Map sum spectrum of (**j**) PbSe-T30 and (**k**) I_2_-PbSe-T30.

**Figure 6 nanomaterials-12-01391-f006:**
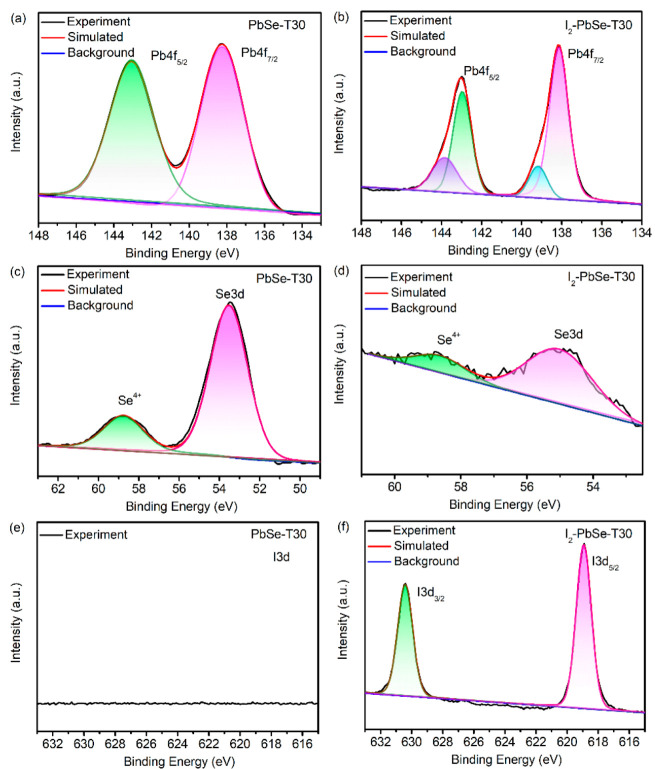
XPS spectra of the PbSe-T30 thin films and I_2_-PbSe-T30: (**a**) Pb 4f of PbSe-T30; (**b**) Pb 4f of I_2_-PbSe-T30; (**c**) Se 3d of PbSe-T30 (**d**) Se 3d of I_2_-PbSe-T30; (**e**) I 3d of PbSe-T30; (**f**) I 3d of I_2_-PbSe-T30.

**Figure 7 nanomaterials-12-01391-f007:**
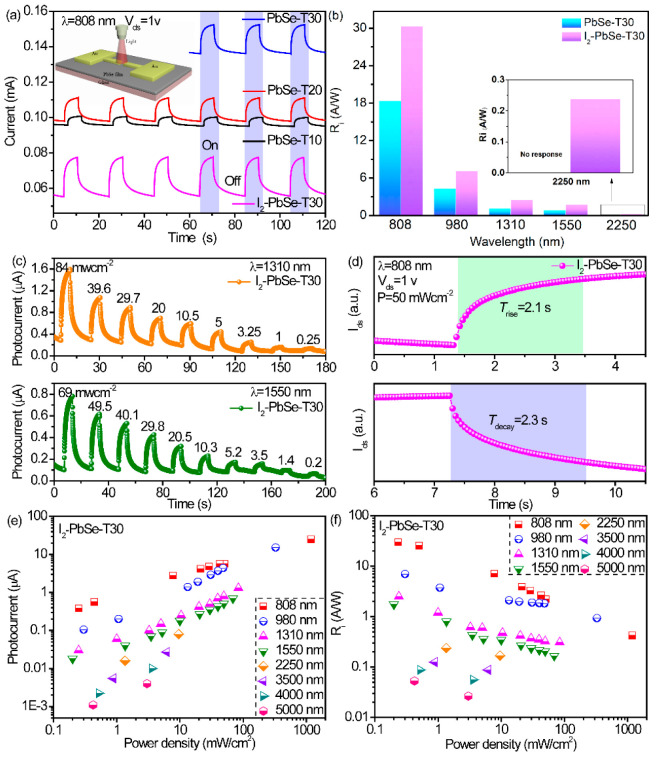
Optoelectronic properties of the PbSe thin film photodetectors: (**a**) Laser on and off photocurrent as a function of time for PbSe-T10, PbSe-T20, PbSe-T30 and I_2_-PbSe-T30 at 1 V under 808 nm with an incident optical power of 1.18Wcm^−2^; (**b**) Comparison of responsivity between PbSe-T30 and I_2_-PbSe-T30 from 808 nm to 2250 nm; (**c**) Photo-current curves of I_2_-PbSe-T30 at and 1330 nm and 1550 nm under different incident optical power; (**d**) The rise time and decay time of photocurrent at 808 nm with an incident optical power of 50 mWcm^−2^ under a bias of 1 V; (**e**) The variation trend of photocurrent with optical power density at 808–5000 nm for I_2_-PbSe-T30; (**f**) Correlation of responsivity of I_2_-PbSe-T30 with optical power density from 808 to 5000 nm.

**Figure 8 nanomaterials-12-01391-f008:**
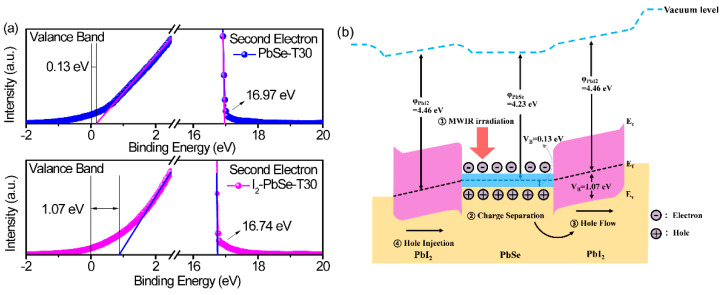
(**a**) Ultraviolet photoelectron spectra of PbSe-T30 and I_2_-PbSe-T30 for work function and valence band edge; (**b**) Band diagram for the I_2_-PbSe-T30 films.

**Table 1 nanomaterials-12-01391-t001:** The EDX comparison of PbSe-T30 and I_2_-PbSe-T30.

Sample	Element	Atomic %	Total
PbSe-T30	Pb	47.15	100%
	Se	52.85	
I_2_-PbSe-T30	Pb	54.9	100%
	Se	29.27	
	I	15.83	

**Table 2 nanomaterials-12-01391-t002:** Comparison preparation and optoelectronic parameters with previous related PbSe photodetector.

Active materials	Preparation	Responsivity [A/W]	Rise/Fall Time	Wavelength [nm]	Ref.
3D PbSe superlattices	Self-assembly method			830	[49]
PbSe crystals-graphene	ECALE	36	1.87/1.8 ms	2700	[8]
PbSe QDs	Wet chemical method	0.064	130/170 ms	980	[50]
PbSe QDs-graphene	Wet chemical method	10^5^	12/49 s	808	[51]
PbSe QDs-MoS_2_	Wet chemical method	1.9 × 10^−6^	250/430 ms	˃1200	[52]
PbSe films-graphene	PVD	0.067	50/150 ms	1070	[22]
PbSe films	PVD	6.6 × 10^−6^	25/50 ms	1070	[39]
PbSe film	Wet chemical method		3 s/5.8 s	1064	[48]
PbSe film	CBD	0.16		2700	[45]
PbSe film	CBD	30.27	2.1/2.3 s	808	This work
PbSe film	CBD	0.053	1.7/1.9 s	5000	This work

## Data Availability

Not applicable.

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
