# Peer review of "Promoted Mid-Infrared Photodetection of PbSe Film by Iodine Sensitization Based on Chemical Bath Deposition"

_nanomaterials, 2022, doi:10.3390/nano12091391_

Round 1

Reviewer 1 Report

review for Promoted Mid-infrared Photodetection of PbSe Film by Iodine Sensitization Based on Chemical Bath Deposition

In this paper Peng et al grow PbSe film usng CBD growth,  characterize the film with optical spectroscopy, XRD, SEM, XPS and finally conduct photoconductive measurements

overall the paper is clear and except XPS part supported by data, but i do not understand the motivation and originality of the paper
for Sure iodine treated PbSe film for mid IR conduction is a well known topic with large litterature associated,  and for sure in nanomaterial i do not expect a revolution, but the authors need to stress what is the novelty of this work

i suggest the author read this recent review on the topic to identify novelty : Mater. Adv., 2021, 2, 3133-3160

when speaking about lead oxide you cite PbO2, but PbS is more likely to get formed

regarding CBD growth, mass are given with illusive accuracy (really 10-4 accuracy)
your Se precursor also include S, may you form PbS as well, what does Xps shows about that

define PDA when used

text above figure 2 (grey atoms are lead and so on...) should actually be part of figure 2 caption
maybe move figure 2b-d to Si and increase part a

about figure 3, you say you see feature at 1400 and 1900, ok there are bumps  but are they really connected to sample ? may it be residue of solvent ?
what is the origin of absorption in the mid IR. it is too broad to be organic. may it be intraband. Please comment

you use SEM to quantize cristalitte size, do you see cossistent result using linewidth of XRD peak ?

for EDX data please plot the graph do not copy past the yellow on blue graph
 final material is very non stoichiometric with large lead excess, comment the effect of it ?

to me xps measurements have likely been done because it was available rather than to determine anything, first the conventional way to plot XPS graph is with low BE on the right (though i agree science remains the same)
when you fit xps data there are some contrains such as keeping same linewidth and constant spin orbit coupling value
while from fig 5 a and b linewidth is not the same at all
the Se doublet with low So coupling likely SO <FWHM for you, is fitted with a single gaussian say it

there is numbering problem for figure 5 and 6 (they are mention as fig 6 and 7 in the text)
in figure 5 avoid E  for power of ten
why UPS analysis i not following XPS one. as is we go back and forth between electronic structure and device

Reviewer 2 Report

 In this manuscript, the authors directly synthesized large-area PbSe films composed of crystal particles of different sizes on amorphous glass substrates using CBD method by 
controlling deposition time. This manuscript can be accepted after addressing the following comments

1- lines 74-75, the authors state that "By controlling the deposition time to design the growth of PbSe thin films, and the films  with different thicknesses and different crystal sizes are obtained " but the authors did not use XRD to calculate the structural parameters such as crystallite size, dislocation density, texture coefficient, ...etc, and also did not correlate the performance of the sample with the crystallite size

2- It is important to study the optical performance in the UV/Vis range also to estimate the bandgap values

3- In  the introduction authors need to clearly show the motivation and novelty of this work 

4- Lines 187-188, How you can calculate the crystallite size from SEM image. There is a big difference between crystallite size and particle size or diameter. Crystallite size can be obtained from XRD but SEM images give the dimensions of the shown particles in the SEM image.  Based on XRD, your films are of polycrystalline nature, so the appearance of particles in SEM may be consists of more than one crystallite.

5-  From EDS spectra, why the Pb atomic ratio increased from ~ 47% to ~55% after incorporation I?

6-Some sentences are very long  and the use of English should be improved

Reviewer 3 Report

The paper with title “Promoted Mid-infrared Photodetection of PbSe Film by Iodine Sensitization Based on Chemical Bath Deposition” presents interesting results.

The paper is well written, but some corrections and clarifications are necessary:

  1. In the Abstract: The phrase “In addition, in order to active the mid-infrared detection ability of the PbSe, we use subsequent iodine treatment to improve the crystalline quality, photoconductivity and reduce the noise, high dark current of PbSe film, explored their use for infrared photodetection.” is not clear. Please rephrased it;
  2. In the “Introduction” part: The topic of the paper is not new; therefore, the authors should underline the novelty of their work because is not clear. What is new, what is better in this paper with respect to other results obtained by other researchers?
  3. Page 2, lines 74-76: the sentence is not clear; please rewrite it;
  4. page 2, line 77: please delete the word “film” from “synthesized films are dense film”;
  5. page 2, lines 77-81: please rephrase the sentence because is not clear: In order to improve the photoelectric response and reduce the noise, high dark current of PbSe film device, using a unique post-deposition sensitization step which annealing in an iodine atmosphere, to explore their use for low-cost fabrication of near-infrared photodetectors.”;
  6. page 2, line 82: please define “Ri”;
  7. page 4, figures 2(a), (b), (c) and (d) should have bigger sizes; in this format the data are unclear;
  8. page 6, line 214: please replace “I” with “(e)”;
  9. page 7, lines 248-250: the authors affirmed that: “The reason why the performance of PbSe-T30 sample is better than the two samples may be due to the different surface morphology which is composed of small nanoparticles and the thickness.”. What is the thickness of the samples? How was determined the thickness (please mention the technique)?
  10. page 9, line 289: please correct the number of the figure: is not 5; it is 6.
  11. In the Conclusions part, please underline why your results are better than others obtained until now. What is new?

I recommend minor revision.

Reviewer 4 Report

In this work, Peng et al. use a facile and convenient chemical bath deposition (CBD) growth strategy to synthesize large-area PbSe films composed of PbSe directly on glass substrates. The effects of the deposition time on the structure, morphology, and optical absorption of the deposited PbSe films were thoroughly investigated by x-ray diffraction, scanning electron microscopy, and infrared spectrometer. In addition, in order to active the mid-infrared detection ability of the PbSe, author use subsequent iodine treatment to improve the crystalline quality, photoconductivity and reduce the noise, high dark current of PbSe film, explored their use for infrared photodetection. This work will be interesting for the reader of the journal. It can be accepted in its current form.
